# Prior Networks for Detection of Adversarial Attacks

## Abstract

Adversarial examples are considered a serious issue for safety critical applications of AI, such as finance, autonomous vehicle control and medicinal applications. Though significant work has resulted in increased robustness of systems to these attacks, systems are still vulnerable to well-crafted attacks. To address this problem several adversarial attack detection methods have been proposed. However, system can still be vulnerable to adversarial samples that are designed to specifically evade these detection methods. One recent detection scheme that has shown good performance is based on uncertainty estimates derived from Monte-Carlo dropout ensembles. Prior Networks, a new method of estimating predictive uncertainty, have been shown to outperform Monte-Carlo dropout on a range of tasks. One of the advantages of this approach is that the behaviour of a Prior Network can be explicitly tuned to, for example, predict high uncertainty in regions where there are no training data samples. In this work Prior Networks are applied to adversarial attack detection using measures of uncertainty in a similar fashion to Monte-Carlo Dropout. Detection based on measures of uncertainty derived from DNNs and Monte-Carlo dropout ensembles are used as a baseline. Prior Networks are shown to significantly out-perform these baseline approaches over a range of adversarial attacks in both detection of whitebox and blackbox configurations. Even when the adversarial attacks are constructed with full knowledge of the detection mechanism, it is shown to be highly challenging to successfully generate an adversarial sample.

## 1 Introduction

Neural Networks (NNs) have become the dominant approach to addressing computer vision (CV) (Girshick, 2015; Simonyan & Zisserman, 2015; Villegas et al., 2017), natural language processing (NLP) (Mikolov et al., 2013b;a; 2010), speech recognition (ASR) (Hinton et al., 2012; Hannun et al., 2014) and bio-informatics (Caruana et al., 2015; Alipanahi et al., 2015) tasks. However, as observed by (Szegedy et al., 2013), they are susceptible to *adversarial attacks* - small perturbations to the input which are almost imperceptible to humans, yet which drastically affect the predictions of the neural network. It was found that adversarial attacks have several properties which make them a serious security concern. Firstly, adversarial attacks are *transferable* - an adversarial attack computed on one network may be able to successfully attack a *different* network (Szegedy et al., 2013; Goodfellow et al., 2015). Secondly, there exists a plethora of adversarial attacks which are quite easy to construct (Goodfellow et al., 2015; Kurakin et al., 2016; Dong et al., 2018; Carlini & Wagner, 2016; Papernot et al., 2016a;b; Liu et al., 2016). Thirdly, it is possible to craft adversarial attacks within the physical world, such as putting stickers on paper (Kurakin et al., 2016). To compound the issue, it was found that adversarial attacks are hard to defend against (Carlini & Wagner, 2017). Specifically, it was found that while adversarial training (Szegedy et al., 2013) and adversarial distillation (Papernot et al., 2016c) are able to improve the robustness of a network, it is still possible to craft successful adversarial attacks against these networks. Indeed, currently, there are far more methods of successfully attacking networks than there are of defending networks(Carlini & Wagner, 2017; 2016). Altogether, this raises serious concerns about how safe it is to deploy neural networks for high-stakes applications, such as autonomous vehicle control, financial and medical applications.

While much work focuses on constructing neural networks which are robust to adversarial attacks, (Carlini & Wagner, 2017) investigate *detection* of adversarial attack and shows that adversarial attacks can be detectable using a range of approaches. However, it turns out that attacks can then be crafted

to fool the proposed detection schemes. However, (Carlini & Wagner, 2017) singles out detection of adversarial attacks using uncertainty measures derived from Monte-Carlo dropout as being successful. Detection of adversarial attack using Monte-Carlo dropout was further investigated in (Smith & Gal, 2018). Work by (Smith & Gal, 2018) suggests that adversarially modified inputs can be interpreted as inputs which lie off the manifold of natural images. In other words, adversarial samples can be seen as a special type of out-of-domain or out-of-distribution input which can be detected by measures of uncertainty derived via Monte-Carlo Dropout.

Recently, (Malinin & Gales, 2018) proposed *Prior Networks* - a new approach to modelling uncertainty which has been shown to outperform Monte-Carlo dropout on a range of tasks. Prior Network parameterize a distribution over output distributions which allows them to separately model *data uncertainty* and *distributional uncertainty*. Furthermore, they are *explicitly* taught to learn decision boundaries between an in-domain and an out-of-domain region, unlike Monte-Carlo dropout, where it is not possible to specify the behaviour of measures of uncertainty in different regions of the input space explicitly. Finally, Prior Networks allow measures of uncertainty to be obtained analytically, without the need for expensive Monte-Carlo sampling. Thus, this work investigates the detection of adversarial attacks using measures of uncertainty derived from Prior Networks. The contributions of this work are as follows:

- Measures for assessing the success of adversarial attacks in the context of detection are proposed.
- It is shown that whitebox and blackbox adversarial attacks with no knowledge of the detection scheme can be successfully detected based on measures of uncertainty.
- It is shown that whitebox detection evading adversarial attacks are difficult to construct for Prior Networks, and detection evading attack fail entirely against Prior Networks.

## 2 UNCERTAINTY FOR DEEP LEARNING

In (Smith & Gal, 2018) adversarially modified inputs are interpreted as inputs which lie off the manifold of natural images - the stronger the adversarial perturbation, the further is the input from the manifold. Thus, adversarial samples can be seen as out-of-distribution inputs. Out-of-distribution inputs are associated with *distributional uncertainty*, which arises due to mismatch between the training and test distributions - in other words *distributional uncertainty* arises when the test data is 'out-of-distribution' relative to the training data. Standard DNNs model *data uncertainty* (uncertainty due to class overlap and noise in the data), as described in appendix B, but fail to capture *distributional uncertainty*. In order to capture *distributional uncertainty* it is necessary to use approaches such as Monte-Carlo dropout or Prior Networks. Thus, this section describes how Monte-Carlo Dropout (Gal & Ghahramani, 2016) and Prior Networks (Malinin & Gales, 2018) model *distributional uncertainty*.

### 2.1 UNCERTAINTY ESTIMATION VIA MONTE-CARLO DROPOUT

The essence of Bayesian approaches is to treat model parameters $\boldsymbol{\theta}$ as random variables and place a prior distribution $p(\boldsymbol{\theta})$ over them to compute a posterior distribution $p(\boldsymbol{\theta}|\mathcal{D})$ over model parameters given the training data $\mathcal{D}$ via Bayes' rule. Uncertainty in the model parameters induces a *distribution over predictive distributions* $P(y|\boldsymbol{x}^*, \boldsymbol{\theta})$ for each observation $\boldsymbol{x}^*$ - each set of model parameters parameterizes a conditional distribution over class labels. The expected distribution $P(y|\boldsymbol{x}^*, \mathcal{D})$ is obtained by marginalizing out the parameters:

$$P(y|\boldsymbol{x}^*, \mathcal{D}) = \int P(y|\boldsymbol{x}^*, \boldsymbol{\theta}) p(\boldsymbol{\theta}|\mathcal{D}) d\boldsymbol{\theta} \tag{1}$$

Unfortunately, both the integral in eq. 1 and calculation of the model posterior are intractable for neural networks. Typically the model posterior distribution is approximated using either an implicit or explicit variational approximation $q(\boldsymbol{\theta})$ and the integral is approximated via sampling (eq. 2), using approaches such as Monte-Carlo dropout (Gal & Ghahramani, 2016):

$$P(y|\boldsymbol{x}^*, \mathcal{D}) \approx \frac{1}{M} \sum_{i=1}^{M} P(y|\boldsymbol{x}^*, \boldsymbol{\theta}^{(i)}), \ \boldsymbol{\theta}^{(i)} \sim q(\boldsymbol{\theta}) \tag{2}$$

Bayesian approaches model *distributional uncertainty* through *model uncertainty*. By selecting an appropriate approximate inference scheme and model prior $p(\boldsymbol{\theta})$ Bayesian approaches aim to craft a model posterior $p(\boldsymbol{\theta}|\mathcal{D})$ such that the ensemble $\{P(\omega_c|\boldsymbol{x}^*, \boldsymbol{\theta}^{(i)})\}_{i=1}^M$ is consistent in-domain and becomes increasingly diverse the further away $\boldsymbol{x}^*$ is from the region of training data. The entropy of the expected distribution $P(\omega_c|\boldsymbol{x}^*, \mathcal{D})$ will indicate the total uncertainty in predictions. Measures of the diversity of the ensemble, such as Mutual Information, assess uncertainty in predictions due to *model uncertainty*, which yields *distributional uncertainty*.

$$\underbrace{\mathcal{MI}[y, \boldsymbol{\theta}|\boldsymbol{x}^*, \mathcal{D}]}_{Model\ Uncertainty} = \underbrace{\mathcal{H}[\mathrm{E}_{p(\boldsymbol{\theta}|\mathcal{D})}[P(y|\boldsymbol{x}^*, \boldsymbol{\theta})]]}_{Total\ Uncertainty} - \underbrace{\mathrm{E}_{p(\boldsymbol{\theta}|\mathcal{D})}[\mathcal{H}[P(y|\boldsymbol{x}^*, \boldsymbol{\theta})]]}_{Expected\ Data\ Uncertainty} \tag{3}$$

In practice, however, for deep, distributed models with tens of million parameters, such as DNNs, it is difficult to select an appropriate approximate inference scheme to craft a model posterior which induces a distribution over distributions with the desired properties.

## 2.2 Uncertainty estimation via Prior Networks

Unlike Bayesian approaches, which indirectly specify a conditional distribution over output distributions, a Prior Network $p(\boldsymbol{\pi}|\boldsymbol{x}^*; \hat{\boldsymbol{\theta}})$, proposed by (Malinin & Gales, 2018) directly parametrizes a prior distribution over categorical output distributions. In this work the Dirichlet distribution (eqn 5) is chosen due to its tractable analytic properties.

$$p(\boldsymbol{\pi}|\boldsymbol{x}^*; \hat{\boldsymbol{\theta}}) = \mathtt{Dir}(\boldsymbol{\pi}|\boldsymbol{\alpha})$$
$$\boldsymbol{\alpha} = \boldsymbol{f}(\boldsymbol{x}^*; \hat{\boldsymbol{\theta}}) \tag{4}$$

A Dirichlet distribution is parameterized by its concentration parameters $\boldsymbol{\alpha}$, where $\alpha_0$, the sum of all $\alpha_c$, is called the *precision* of the Dirichlet distribution. Higher values of $\alpha_0$ lead to sharper, more confident distributions:

$$\mathtt{Dir}(\boldsymbol{\pi}|\boldsymbol{\alpha}) = \frac{\Gamma(\alpha_0)}{\prod_c^K \Gamma(\alpha_c)} \prod_{c=1}^K \pi_c^{\alpha_c - 1}, \quad \alpha_c > 0, \ \alpha_0 = \sum_c^K \alpha_c \tag{5}$$

The predictive distribution is given by the expected categorical distribution under the Dirichlet prior:

$$P(\omega_c|\boldsymbol{x}^*; \hat{\boldsymbol{\theta}}) = \int P(\omega_c|\boldsymbol{\pi}) p(\boldsymbol{\pi}|\boldsymbol{x}^*; \hat{\boldsymbol{\theta}}) d\boldsymbol{\pi} = \frac{\alpha_c}{\alpha_0} \tag{6}$$

The desired behaviors of the Prior Network can be visualized on a simplex (fig 1), where figure 1:a describes confident behavior (low-entropy prior focused on low-entropy output distributions), figure 1:b describes uncertainty due severe class overlap and figure 1:c describes the behaviour for an out-of-distribution input.

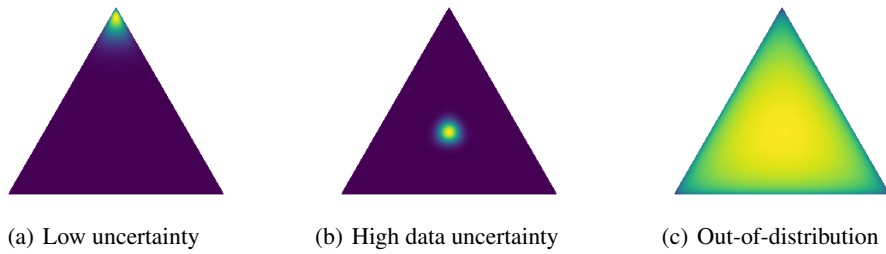

(a) Low uncertainty        (b) High data uncertainty        (c) Out-of-distribution

Figure 1: Desired Behaviors of a Dirichlet distribution over categorical distributions.

A Prior Network is trained to display these behaviors by minimizing the following multi-task loss functions, as described in (Malinin & Gales, 2018). A flat Dirichlet is chosen as the out-of-distribution target distribution $p_{\mathtt{out}}(\boldsymbol{\pi})$, as it is the maximum entropy Dirichlet distribution. The target in-domain distribution $p_{\mathtt{in}}(\boldsymbol{\pi}|\boldsymbol{x})$ is constructed by specifying the mean and precision of the Dirichlet, as

described in (Malinin & Gales, 2018). In order to train with this loss function, it is necessary to have out-of-distribution training data. One way of doing this is to chose, in addition to the target training data, an 'out-of-distribution' dataset. For example, if a model is trained on CIFAR-10 (Krizhevsky, 2009), it is possible to use CIFAR-100 as the out-of-distribution dataset, as they don't have overlapping classes.

$$\mathcal{L}(\boldsymbol{\theta}) = \mathrm{E}_{\mathrm{p}_{\mathrm{in}}(\boldsymbol{x})}\big[D_{KL}[\mathrm{p}_{\mathrm{in}}(\boldsymbol{\pi}|\boldsymbol{x})||\mathrm{p}(\boldsymbol{\pi}|\boldsymbol{x};\boldsymbol{\theta})]\big] + \mathrm{E}_{\mathrm{p}_{\mathrm{out}}(\boldsymbol{x})}\big[D_{KL}[\mathrm{p}_{\mathrm{out}}(\boldsymbol{\pi})||\mathrm{p}(\boldsymbol{\pi}|\boldsymbol{x};\boldsymbol{\theta})]\big] \qquad (7)$$

Given a trained Prior Network it is possible to calculate the Mutual Information using an expression similar to equation 3:

$$\underbrace{\mathcal{MI}[y,\boldsymbol{\pi}|\boldsymbol{x}^*;\hat{\boldsymbol{\theta}}]}_{Distributional\ Uncertainty} = \underbrace{\mathcal{H}[\mathrm{E}_{\mathrm{p}(\boldsymbol{\pi}|\boldsymbol{x}^*;\hat{\boldsymbol{\theta}})}[\mathrm{P}(y|\boldsymbol{\pi})]]}_{Total\ Uncertainty} - \underbrace{\mathrm{E}_{\mathrm{p}(\boldsymbol{\pi}|\boldsymbol{x}^*;\hat{\boldsymbol{\theta}})}[\mathcal{H}[\mathrm{P}(y|\boldsymbol{\pi})]]}_{Expected\ Data\ Uncertainty} \qquad (8)$$

This expression, as in the Bayesian case, allows total uncertainty, given by the entropy of the expected distribution, to be decomposed into *data uncertainty* and *distributional uncertainty*

## 3 MEASURES OF PERFORMANCE

In order to investigate detection of adversarial attacks, it is necessary to discuss how to assess the effectiveness of an adversarial attack in a scenario where detection of the attack is possible. Previous work on detection of adversarial examples (Gong et al., 2017; Grosse et al., 2017; Metzen et al., 2017; Carlini & Wagner, 2017; Smith & Gal, 2018) assesses the performance of detection methods separately from whether an adversarial attack was successful, and use the standard measures of adversarial success and detection performance. However, in a real deployment scenario, an attack can only be considered successful if it *both* affects the predictions *and* evades detection. In this section we develop a measure of performance to assess this.

An adversarial input $\boldsymbol{x}_{\mathrm{adv}}$ will be defined as the output of an adversarial attack generation process $\mathcal{A}_{\mathrm{adv}}$ applied to a *natural* input $\boldsymbol{x}$:

$$\mathcal{A}_{\mathrm{adv}}(\boldsymbol{x},t) = \arg\min_{\tilde{\boldsymbol{x}}\in\mathcal{R}^K}\Big\{\mathcal{L}\big(\mathrm{P}(y|\tilde{\boldsymbol{x}};\hat{\boldsymbol{\theta}}),t\big) + \delta(\boldsymbol{x},\tilde{\boldsymbol{x}})\Big\} \qquad (9)$$

This process is typically an optimization problem which tries to minimize a loss $\mathcal{L}(\cdot)$, which is a function of the input $\boldsymbol{x}$, the model $\mathrm{P}(y|\tilde{\boldsymbol{x}};\hat{\boldsymbol{\theta}})$ and a target class $t$, with respect to the input while also minimizing some distance $\delta(\cdot,\cdot)$ between the original image and the generated adversarial perturbation. This loss is typically the negative log-likelihood of a particular target class, in case of targeted adversarial attacks, or the (positive) log-likelihood of the predicted class, in case of an untargeted attack:

$$\begin{aligned}\mathcal{L}_{tgt}\big(\mathrm{P}(y|\tilde{\boldsymbol{x}};\hat{\boldsymbol{\theta}}),t\big) &= -\ln\mathrm{P}(\omega_t|\tilde{\boldsymbol{x}};\hat{\boldsymbol{\theta}}) \\ \mathcal{L}_{utgt}\big(\mathrm{P}(y|\tilde{\boldsymbol{x}};\hat{\boldsymbol{\theta}}),t\big) &= \ln\mathrm{P}(\omega_t|\tilde{\boldsymbol{x}};\hat{\boldsymbol{\theta}}), \quad \omega_t = \arg\max_{\omega}\{\mathrm{P}(y=\omega|\boldsymbol{x};\hat{\boldsymbol{\theta}})\}\end{aligned} \qquad (10)$$

The distance $\delta(\cdot,\cdot)$, typically the $L_1$, $L_2$ or $L_\infty$ norm, is minimized so that the adversarial attack is still *perceived* to be a natural input to a human observer. The best adversarial attack is one which minimizes the chosen loss and has the minimal deviation from the original image $\boldsymbol{x}$. There are multiple ways in which this optimization problem can be solved (Szegedy et al., 2013; Goodfellow et al., 2015; Kurakin et al., 2016; Dong et al., 2018).

While the process $\mathcal{A}_{\mathrm{adv}}$ will always generate some kind of perturbation, it will not always yield a successful attack, as it may fail to affect the prediction. For the purposes of this discussion the adversarial generation process $\mathcal{A}_{\mathrm{adv}}$ will be defined to either yield a successful adversarial attack $\boldsymbol{x}_{\mathrm{adv}}$ or an empty set $\emptyset$.

$$\mathcal{A}_{\mathrm{adv}}(\boldsymbol{x},t) \in \{\boldsymbol{x}_{\mathrm{adv}},\emptyset\} \qquad (11)$$

It is necessary to point out that here we take the perspective of *the attacker* - a successful attack, in the case of targeted attacks, is one which yields *the target class*. However, consider the perspective of the system designer who does not know whether the attacker attempts a targeted attack or not. From the point of view of the system designer, any attack which is able to affect the predictions of the network

may be considered successful, regardless of whether the attack produced the attacker's desired prediction. In the case of non-targeted attacks, however, the attacker's and defender's definition of success are the same.

In a standard scenario, where there is no detection, the efficacy of an adversarial attack on a model[1] can be summarized via the *success rate* $S$ of the attack:

$$S = \frac{1}{N} \sum_{i=1}^{N} \mathcal{I}(\mathcal{A}_{\text{adv}}(\boldsymbol{x}_i, t)), \quad \mathcal{I}(\boldsymbol{x}) = \begin{cases} 1, & \boldsymbol{x} \neq \emptyset \\ 0, & \boldsymbol{x} = \emptyset \end{cases} \tag{12}$$

Typically $S$ is plotted against the total maximum perturbation $\epsilon$ from the original image, measured as either the $L_1$, $L_2$ or $L_\infty$ distance from the original image.

Now consider using a threshold-based detection scheme where a sample is labelled 'positive' if some measure of uncertainty, such as entropy $\mathcal{H}(\boldsymbol{x})$, is less than a threshold $T$ and 'negative' if it is higher than a threshold:

$$\mathcal{I}_T(\boldsymbol{x}) = \begin{cases} 1, & T > \mathcal{H}(\boldsymbol{x}) \\ 0, & T \leq \mathcal{H}(\boldsymbol{x}) \end{cases} \tag{13}$$

The performance of such a scheme can be evaluated at every threshold value using the *true positive rate* $t_p(T)$ and the *false positive rate* $f_p(T)$:

$$t_p(T) = \frac{1}{N} \sum_{i=1}^{N} \mathcal{I}_T(\boldsymbol{x}_i) \qquad f_p(T) = \frac{1}{N} \sum_{i=1}^{N} \mathcal{I}_T(\mathcal{A}_{\text{adv}}(\boldsymbol{x}_i, t)) \tag{14}$$

The whole range of such trade offs can be visualized using a Receiver-Operating-Characteristic (ROC) and the quality of the trade-off can be summarized using area under the ROC curve.

However, a standard ROC curve does not give credit to the system for being robust to adversarial attacks - it doesn't account for situations where the process $\mathcal{A}_{\text{adv}}(\cdot)$ fails to produce a successful attack. In fact, if an adversarial attack is made against a system which has a detection scheme, it can only be considered successful if it *both* affects the predictions *and* evades detection. This condition can be summarized in the following indicator function:

$$\hat{\mathcal{I}}_T(\boldsymbol{x}) = \begin{cases} 1, & T > \mathcal{H}(\boldsymbol{x}) \\ 0, & T \leq \mathcal{H}(\boldsymbol{x}) \\ 0, & \boldsymbol{x} = \emptyset \end{cases} \tag{15}$$

Given this indicator function, a new false positive rate $\hat{f}_P(T)$ can be defined as:

$$\hat{f}_p(T) = \frac{1}{N} \sum_{i=1}^{N} \hat{\mathcal{I}}_T(\mathcal{A}_{\text{adv}}(\boldsymbol{x_i}, t)) \tag{16}$$

This false positive rate can now be seen as a new *Joint Success Rate* which measures how many attacks were both successfully generated and evaded detection, given the threshold of the detection scheme. The *Joint Success Rate* can be plotted against the standard true positive rate on an ROC curve to visualize the possible trade-offs. This ROC curve will now give credit to the system when the attacker fails to generate a succesful attack. One possible operating point is where the false positive rate is equal to the false negative rate, also known as the *Equal Error-Rate* point:

$$\hat{f}_P(T_{\text{EER}}) = 1 - t_P(T_{\text{EER}}) \tag{17}$$

Throughout the remainder of this work the EER false positive rate will be quoted as the *Joint Success Rate*.

## 4 DETECTING ADVERSARIAL ATTACKS

The previous sections discussed how to obtain estimates of uncertainty using several different approaches and how to assess the performance of adversarial attacks in the scenario where a detection scheme is in place. In this section detection of adversarial samples using measures of uncertainty is investigated. Four threat models are evaluated:

---

[1]Given an evaluation dataset $\mathcal{D}_{\text{test}} = \{\boldsymbol{x}_i, y_i\}_{i=1}^{N}$

- Whitebox attack with no knowledge of the detection scheme;
- Blackbox attack with no knowledge of the detection scheme;
- Whitebox attack with complete information about the detection scheme;
- Blackbox attack with complete information about the detection scheme.

Experiments were conducted on the CIFAR-10 (Krizhevsky, 2009) dataset. Four models were constructed: standard classification network DNN, Monte-Carlo dropout (MCDP) ensemble derived from the same DNN, a 'standard' prior network 'PN' and an 'adversarially trained' prior network 'PN-ADV'. DNN and MCDP were used as baselines in this work. The baseline prior network PN was trained on CIFAR-10 as in-domain data and CIFAR-100 as out-of-distribution data in exactly the same setup as (Malinin & Gales, 2018). The adversarially trained Prior Network PN-ADV was additionally trained on un-targeted Fast Gradient Sign Method (Goodfellow et al., 2015) attacks which were dynamically generated during training. The perturbation values $\epsilon$ were randomly sampled from a normal distribution for each mini-batch. Standard VGG-16 (Simonyan & Zisserman, 2015) architecture was used for all networks. Entropy of the predictive distribution was used as the measure of uncertainty of a DNN and Mutual Information was used as the measure of uncertainty for MCDP (eqn 3) and Prior Networks (eqn 8). For each model type, 5 model-instantiations with different random seeds were trained.

Three types of non-targeted adversarial attacks are considered: Fast-Gradient Sign Method (FGSM) attacks (Szegedy et al., 2013), Basic-Iterative Method (BIM) Attacks (Kurakin et al., 2016) and Momentum-Iterative Method (MIM) (Dong et al., 2018) attacks. These attacks are chosen because they were the ones evaluated in (Smith & Gal, 2018) and MIM attacks are considered to be quite strong. The $L_\infty$ norm version of the FGSM, BIM and MIM attacks were investigated. Each of the iterative attacks was run for 10 iterations of gradient descent with their 'default' settings , as described in (Kurakin et al., 2016; Dong et al., 2018). Further details on the training and construction of all models can be found in appendix A.

### 4.1 ATTACKS WITH NO KNOWLEDGE ABOUT THE DETECTION SCHEME

Figure 2 displays plots of the *Joint Success Rate* (eqn 16) at the EER threshold against the perturbation $\epsilon$ for all attacks averaged across different seed values with $\pm\sigma$ bounds.

Figures 2a-c show that a deterministic DNN and MC Dropout achieve nearly identical performance across all attacks. Curiously, they are both able to detect FGSM attacks, but are almost completely ineffective against BIM and MIM attacks, which achieve joint success rates in excess of 90%[2] for small values of epsilon. The standard prior network PN is robust FGSM attacks and is more robust against BIM and MIM attacks than the baselines - the highest success rate of BIM and MIM attack is now 80% at a higher epsilon value of 30. The adversarially trained prior network PN-ADV achieves the best performance by a large margin across all attacks. It is totally robust to FGSM attacks and significantly more robust to BIM and MIM attacks, which now achieve a peak joint success rate of $\tilde{1}5\%$ in a narrow range of perturbations.

Performance against black-box attacks is described in figures 2d-f. Whitebox attacks against each random seed are used as blackbox attacks against all the other seeds for a model type. All models, even the DNN and MCDN, are able to achieve very low joint success rates and are robust to all attacks considered. Curiously, a standard Prior Network PN has the worst performance. However, an adversarially trained prior network achieves by far the best performance against all considered black-box adversarial attacks.

These results indicate that a standard prior network learns a better decision boundary between the in-domain and out-of-domain regions than Monte-Carlo dropout and is able to yield better measures of uncertainty. However, as it was trained using only 'on-manifold' out-of-distribution data, it still is susceptible to adversarial attacks which are far from the manifold. Training a prior network on FGSM data allows it to learn a better decision boundary, allowing the prior network to generalize the out-of-domain region beyond the manifold of natural data. Interestingly, the results suggest that the

---

[2]Note - a joint success rate larger than 50% means that the AUC is less than 0.5 and can be improved by using a reversed detection scheme

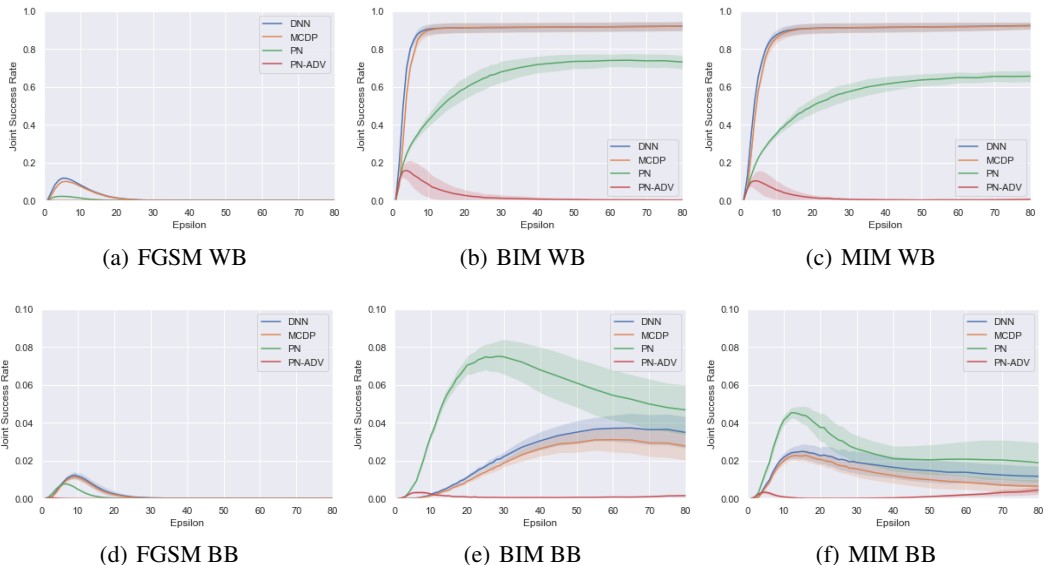

Figure 2: Plots of joint success rate at EER threshold for whitebox (a-c) and blackbox (d-f) attacks. **Note difference in y-axis scale of plots d-f**.

'holes' in a model's understanding of where the in-domain/out-of-domain region discovered using whitebox attacks fail to generalize as blackbox attacks.

## 4.2 ATTACKS WITH COMPLETE KNOWLEDGE ABOUT DETECTION SCHEME

The results of section 4.1 show that prior networks are able to detect both whitebox and blackbox adversarial attacks successfully. However, as discussed in (Carlini & Wagner, 2017), it is necessary to evaluate whether is is possible to construct adversarial attacks specifically to evade detection. This section investigate whether is is possible craft adversarial attacks which avoid detection by measures of uncertainty derived from DNNs, Monte-Carlo Dropout and Prior Networks.

Conceptually, the best approach to avoiding detection using measures of uncertainty is to generate an adversarial attack which changes a model's prediction while leaving the measures of uncertainty (entropy, mutual information) unchanged. In the case of a DNN or Monte-Carlo dropout, the simplest approach to do this is to simply permute the predicted distribution over classes so that the probability of the max class is assigned to the target class, and the probability of the target class is assigned to the max class. The loss function minimized by the adversarial generation process will be the KL divergence between the predicted distribution over class labels $P(y|\tilde{\boldsymbol{x}}; \hat{\boldsymbol{\theta}})$ and the target permuted distribution $P_{\mathtt{t}}(y)$: $\mathcal{A}_{\mathtt{adv}}$ applied to a *natural* input $\boldsymbol{x}$:

$$\mathcal{L}\big(P(y|\tilde{\boldsymbol{x}}; \hat{\boldsymbol{\theta}}), t\big) = D_{KL}(P_{\mathtt{t}}(y)||P(y|\tilde{\boldsymbol{x}}; \hat{\boldsymbol{\theta}})) \tag{18}$$

For prior networks the equivalent approach would be to permute the values of $\boldsymbol{\alpha}$ and to minimize KL divergence to the permuted target Dirichlet distribution:

$$\mathcal{L}\big(P(y|\tilde{\boldsymbol{x}}; \hat{\boldsymbol{\theta}}), t\big) = D_{KL}(p_{\mathtt{t}}(\boldsymbol{\pi})||p(\boldsymbol{\pi}|\tilde{\boldsymbol{x}}; \hat{\boldsymbol{\theta}})) \tag{19}$$

In the following experiments, MIM adversarial attacks are generated using the loss functions given by equations 18 and 19 for DNN/MCDP and Prior Network models, respectively. The iterative approaches are run for a range of iterations, ranging from 10 to 100. The *second most likely* class is selected as the target class, as this should be a more aggressive attack than switching to the least likely class. Figure 3 shows the *Joint Success Rate*, ROC AUC and success rate $S$ (eqn. 12) vs the number of iterations for MIM attack generation at a perturbation of 40. The change of x-axis from perturbation $\epsilon$ to iteration is used to illustrate the added computational difficulty in generating detection evading attacks.

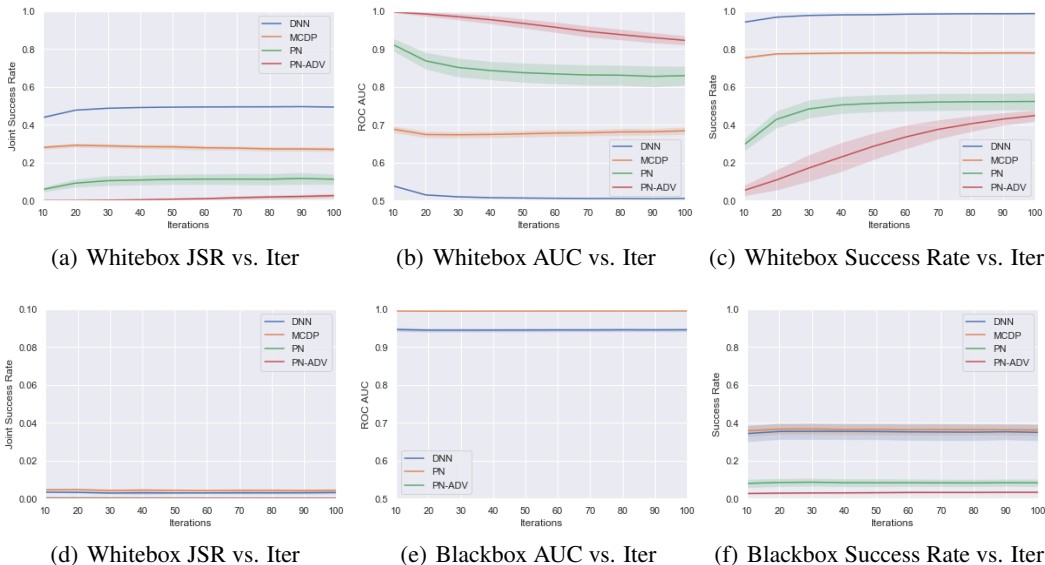

Figure 3: Summaries of performance against whitebox and blackbox detection evading MIM attack.
**Note different y-axis scale of plot d.**

The figures show that a detection evading whitebox attack against a DNN is completely successful.
Figure 3c shows that in almost 100% of cases the attack yields the target class. Monte-Carlo dropout
performs better in terms of success rate and AUC. However, whitebox detection evading attacks
against PN and PN-adv were far less successful, yielding the target class 45% and 40% of the time
after far more iterations, respectively. Furthermore, AUC only fell by about 10% absolute for prior
network models, and and only after many iterations. Interestingly, blackbox detection evading attacks
failed altogether and were detectable using all considered models. This suggests that it is very difficult
to generalize the weaknesses a detection evading attack discovers in a model's detection scheme.

The experiments in this section show that it is non-trivial to successfully construct whitebox adversar-
ial attacks which are able to evade detection by measures of uncertainty for appropriately secured
prior networks. This suggests that the optimization problem in equation 9 becomes more difficult to
solve for Prior Networks, which, in turn, suggests that the manifold hypothesis of adversarial attacks
is correct. Specification of the behaviour of the network in-domain and out-of-domain *on-manifold* as
well as out-of-domain *off-manifold* greatly constrains the space of solutions to equation 9 where the
attack both yields the target class *and* avoids changing properties of predicted distribution over distri-
butions. Using measures of uncertainty in the prediction, uncertainty in the model's understanding of
the data, allows the space of solutions to the adversarial optimization problem to be constrained in a
way which methods proposed in (Metzen et al., 2017; Gong et al., 2017; Grosse et al., 2017) do not.

## 5    CONCLUSION

This work shows that it is possible to detect FGSM, BIM and MIM adversarial attacks using measures
of uncertainty derived from DNNs, MC Dropout and Prior Networks. Prior Network models are the
most robust to FGSM, BIM and MIM $L_\infty$ adversarial attacks and outperform all the other methods
by a large margin in both detection of whitebox and blackbox adversarial attacks. In section 4.2
it is shown that it is possible to construct targeted adversarial attacks which also avoid detection
by directly attacking the measure of uncertainty derived from DNNs and Monte-Carlo dropout
ensembles. However, results in section 4.2 show that it difficult to construct detection avoiding
attacks for adversarially trained prior networks. The results in this work are encouraging and show
that is appropriate measures of uncertainty are used, then adversarial attacks are not a great security
concern. However, further work in evaluating the robustness of uncertainty estimates derived from
Prior Networks and Monte-Carlo dropout against stronger detection-evading adversarial attacks must
be done.

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

## APPENDIX A  DETAILS OF EXPERIMENTAL SET

Models were trained on the CIFAR-10 dataset (Krizhevsky, 2009). Prior Networks were additionally trained on CIFAR-100 data as out-of-distribution training data. Models were implemented in

| Dataset | Train | Test | Classes |
|---------|-------|------|---------|
| CIFAR-10 | 50000 | 10000 | 10 |
| CIFAR-100 | 50000 | 10000 | 100 |

Table 1: Training and Evaluation Datasets

Tensorflow Abadi et al. (2015) using a standard convolutional VGG-16 (Simonyan & Zisserman, 2015) architecture. The only difference is that the fully connected layers have a dimensionality of 2048, rather than 4096, and that Leaky-ReLU activations were used instead of ReLU. The dropout keep probability for the convolution layers was 0.3+ the keep probability for fully connected layer. The input features were scaled to be between -1.0 and 1.0 rather than 0 and 255. Data augmentation was used by flipping the images left-right randomly, shifting images by up to 4 pixels up/down and left-right and adding a random rotation of up to $\pm$ 0.25 radians.

All models are trained using the NADAM optimizer (Dozat, 2016) using a 1-cycle learning rate policy. Learning rates were linearly increased from the initial learning rate to 10x the initial learning rate for half the cycle, and then linearly decreased down to the initial learning rate for the second half cycle. They were then linearly decreased for the remained of the training epochs to a final learning rate of 1e-6. The training configuration for all models is described in table 2.

| Model | Dropout | init LR | Cycle Length | Total Epochs | OOD data |
|-------|---------|---------|--------------|--------------|----------|
| DNN | 0.5 | 1e-3 | 30 | 45 | - |
| PN | 0.7 | 7.5e-4 | 70 | 100 | CIFAR-100 |
| PN-ADV | 0.7 | 7.5e-4 | 70 | 100 | CIFAR-100 + FGSM |

Table 2: Training Configuration

The classification error rates on the test data for all models are given in the table 3, which shows that all models have comparable error rates and that adversarial training has improved the error rate of PN-ADV relative to PN.

| Model | % Error |
|-------|---------|
| DNN | 8.0 +/- 0.3 |
| MCDP | 8.0 +/- 0.3 |
| PN | 8.5 +/- 0.1 |
| PN-ADV | 8.2 +/- 0.1 |

Table 3: Classification Error rates on CIFAR-10 test data.

Table 4 shows the out-of-distribution detection performance using each model. Here, SVHN (Goodfellow et al., 2013), LSUN (Yu et al., 2015) and TinyImageNet (CS231N, 2017) as used as out-of-distribution data and CIFAR-10 test is the in-distribution data. These experiments are run in the same fashion as in (Malinin & Gales, 2018). The results show that using out-of-distribution detection performance is enhanced a little when using untargeted FGSM attacks as additional out-of-distribution training data for PN-ADV.

| Model | SVHN | LSUN | TinyImageNet |
|-------|------|------|--------------|
| DNN | 90.8 +/- 1.3 | 91.4 +/- 0.4 | 88.7 +/- 0.8 |
| MCDP | 83.7 +/- 1.6 | 89.3 +/- 0.3 | 86.9 +/- 0.3 |
| PN | 98.5 +/- 0.2 | 94.6 +/- 0.2 | 94.6 +/- 0.3 |
| PN-ADV | 98.5 +/- 0.2 | 95.1 +/- 0.6 | 94.9 +/- 0.1 |

Table 4: Out-of-Distribution Detection

## APPENDIX B    UNCERTAINTY FOR DNNS

The simplest approach to modeling uncertainty for classification is to use a DNN to parameterize a discrete distribution over class labels conditioned on the input (eq. 20):

$$\mathrm{P}(y|\boldsymbol{x}^*;\boldsymbol{\theta}) = \boldsymbol{f}(\boldsymbol{x}^*;\boldsymbol{\theta}) \tag{20}$$

which is the standard approach to classification using neural networks. This model will capture *data uncertainty* when trained via maximum likelihood. Specifically, consider the derivation in eq. 21, which shows that the expected negative log-likelihood of a model, given the real underlying data distribution $\mathrm{P}_{\mathtt{tr}}(y,\boldsymbol{x})$ can be expressed as the expected KL divergence between the model $\mathrm{P}(y|\boldsymbol{x};\boldsymbol{\theta})^3$ and the true conditional distribution $\mathrm{P}_{\mathtt{tr}}(y|\boldsymbol{x})$, which is the reducible loss, and the entropy of $\mathrm{P}_{\mathtt{tr}}(y|\boldsymbol{x})$, which is the irreducible loss. The irreducible loss represents the *data uncertainty* - uncertainty due to, for example, class overlap. Thus, as the reducible loss is minimized, the model learns not only to yield the correct classifications, but also to capture the uncertainty inherent in the data.

$$
\begin{aligned}
\mathrm{E}[\mathcal{L}_{CE}(\boldsymbol{\theta})] &= \mathrm{E}_{\mathrm{P}_{\mathtt{tr}}(\boldsymbol{x})}\Big[ -\sum_c^K \mathrm{P}_{\mathtt{tr}}(\omega_c|\boldsymbol{x}) \ln \mathrm{P}(\omega_c|\boldsymbol{x};\boldsymbol{\theta}) \Big] \\
&= \mathrm{E}_{\mathrm{P}_{\mathtt{tr}}(\boldsymbol{x})}\Big[ \underbrace{\mathrm{KL}(\mathrm{P}_{\mathtt{tr}}(y|\boldsymbol{x})||\mathrm{P}(y|\boldsymbol{x};\boldsymbol{\theta}))}_{Reducible\ Loss} + \underbrace{\mathcal{H}[\mathrm{P}_{\mathtt{tr}}(y|\boldsymbol{x})]}_{Irreducible\ Loss} \Big]
\end{aligned}
\tag{21}
$$

Thus, it is possible to use the entropy of the posterior over classes (eq. 22) as a *measure of uncertainty*. This was evaluated as a baseline approach to misclassification detection and out-of-distribution sample detection in (Hendrycks & Gimpel, 2016) and (Malinin & Gales, 2018).

$$\mathcal{H}[\mathrm{P}(y|\boldsymbol{x}^*;\boldsymbol{\theta})] = -\sum_{c=1}^K \mathrm{P}(\omega_c|\boldsymbol{x}^*;\boldsymbol{\theta}) \ln \mathrm{P}(\omega_c|\boldsymbol{x}^*;\boldsymbol{\theta}) \tag{22}$$

Unfortunately, DNNs are unable to model *distributional uncertainty* as the behaviour of DNNs for out-of-distribution or off-manifold inputs is unspecified. Thus, they may yield both low-entropy and high-entropy posteriors over class labels in these regions.

## APPENDIX C    PLOTS FOR ALL EXPERIMENTS

### C.1    WHITEBOX ATTACK WITH NO KNOWLEDGE OF DETECTION SCHEME

Plots in figure 4 show a complete summary of all performance metrics for whitebox FGSM, BIM and MIM attacks with no knowledge of the detection scheme. Note, that the AUC for DNN, MCDP and PN for BIM and MIM attacks is less than 0.5, which indicates that the uncertainty for adversarial attacks is lower than for real data. In this 'perverse' situation better performance can be achieved by flipping the detection criterion. However, it be better to define a different detection two-variable criterion based:

$$\mathcal{I}_{\alpha,\beta}(\boldsymbol{x}) = \begin{cases} 1, & |\alpha - \mathcal{H}(\boldsymbol{x})| < \beta \\ 0, & |\alpha - \mathcal{H}(\boldsymbol{x})| \geq \beta \\ 0, & \boldsymbol{x} = \emptyset \end{cases} \tag{23}$$

In this case, any measure of uncertainty which differs by more than $\beta$ from $\alpha$ is considered out-of-distribution. However, it is more difficult to analyze all the trade-off of a two-variable detection criterion.

---

[3] $\mathrm{P}(\omega_c|\boldsymbol{x};\boldsymbol{\theta})$ is a shorthand for $\mathrm{P}(y=\omega_c|\boldsymbol{x};\boldsymbol{\theta})$

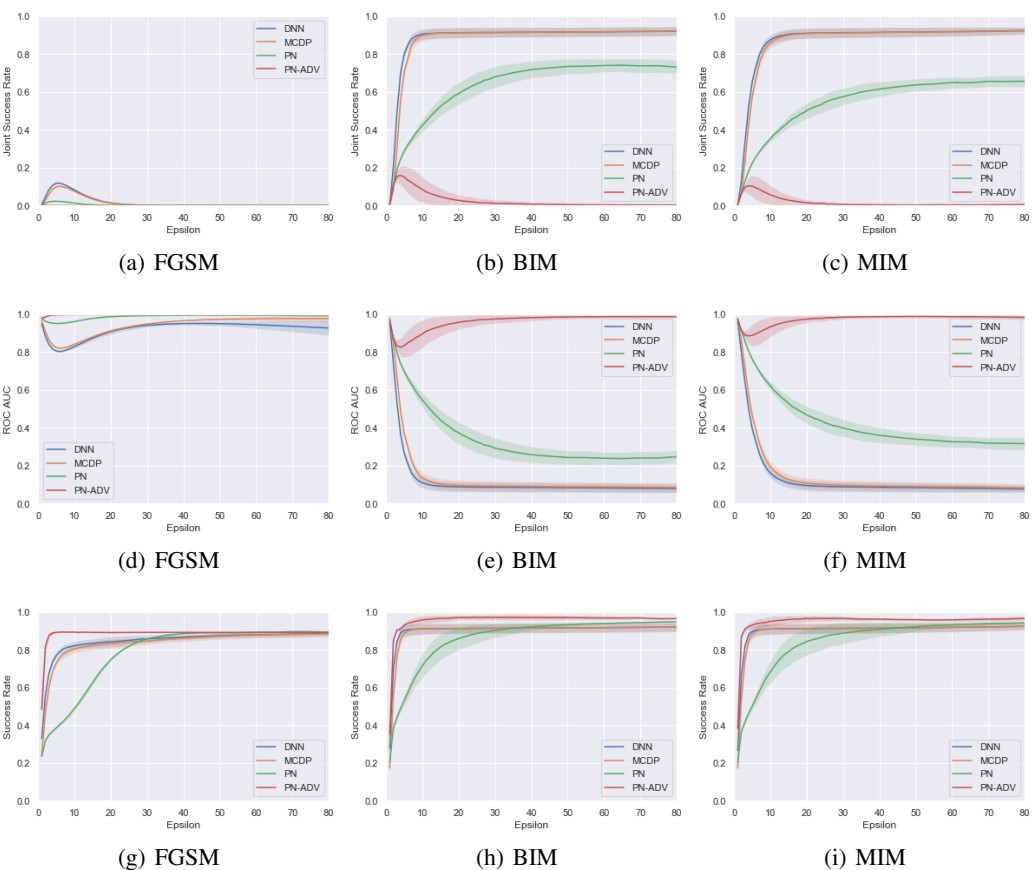

Figure 4: Plots of Joint Success Rates, ROC AUC and Success rates against perturbation ($\epsilon$) for whitebox FGSM, BIM and MIM attacks.

## C.2    BLACKBOX ATTACK WITH NO KNOWLEDGE OF DETECTION SCHEME

(a) FGSM

(b) BIM

(c) MIM

(d) FGSM

(e) BIM

(f) MIM

(g) FGSM

(h) BIM

(i) MIM

Figure 5: Plots of Joint Success Rates, ROC AUC and Success rates against perturbation ($\epsilon$) for whitebox FGSM, BIM and MIM attacks.

Plots in figure 4 show a complete summary of all performance metrics for blackbox FGSM, BIM and MIM attacks with no knowledge of the detection scheme. Note the difference in y-axis scale between figures 4a-c and figures 5a-c.

## C.3 WHITEBOX ATTACK WITH FULL KNOWLEDGE OF DETECTION SCHEME

Figure 6: Plots of ROC Curves of Entropy of Predictive Distribution

Plots in figure 6 show a complete summary of all performance metrics for whitebox BIM and MIM attacks with full knowledge (access to parameters) of the detection scheme.

The distribution of uncertainties (entropy and mutual information) yielded by each model for real data and detection evading adversarial attacks is given in figure 7. These figures show that detection-evading adversarial attacks are able to perfectly preserve the distribution of entropy for a DNN and almost perfectly preserve the Mutual Information of MCDP. However, they have greater difficulty for PN and are unable to match the distribution of Mutual Information for PN-ADV.

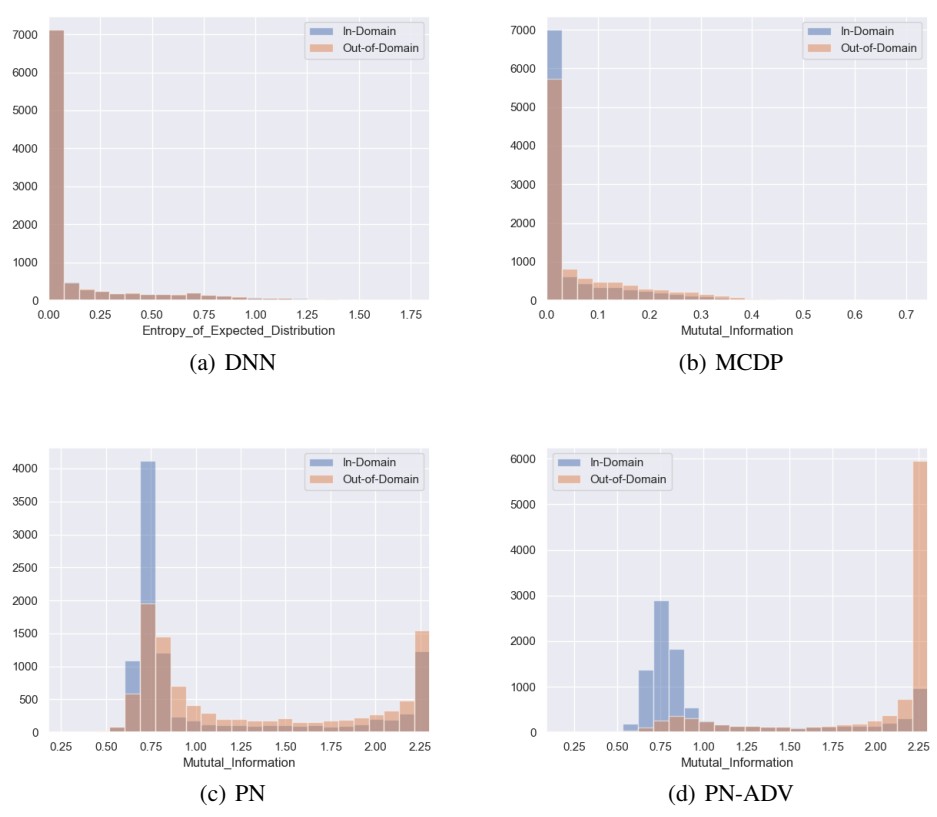

Figure 7: Plots of uncertainty distributions for real (in-domain) and adversarial (out-of-domain) data for MIM attack at iteration 100

## C.4 BLACKBOX ATTACK WITH FULL KNOWLEDGE OF DETECTION SCHEME

Figure 8: Plots of ROC Curves of Entropy of Predictive Distribution

Plots in figure 8 show a complete summary of all performance metrics for blackbox BIM and MIM attacks with full knowledge (access to parameters) of the detection scheme. Note the difference in y-axis scale between figures 6a-b and figures 8a-b.

