# OpenReview forum: "Prior Networks for Detection of Adversarial Attacks"
_ICLR.cc/2019/Conference_

### Official Review · AnonReviewer3 · 2018-11-02
**interesting idea of adversary detection using uncertainty prediction; the analysis is not sufficient to make robust claims and the novelty has to be differentiated from existing work**

**Rating:** 4
**Confidence:** 5

**Review:**

This paper proposed the use of uncertainty measure evaluated by the prior network framework in (Malinin and Gales 2018) to detect adversarial inputs. Empirically, the best detector against three L_infinity based attacks (FGSM, BIM and MIM), is a prior network that is adversarially trained with FGSM, in both white-box and black-box settings. The results also showed superior performance over a detector based on Monte Carlo Dropout methods (MCDP). Although the idea is interesting and the presented results seem promising, there are some key experiments lacking that may prevent this work from making its claims on robustness and detectability. The detailed comments are as follows.

1. Detection performance against high-confidence adversarial examples is lacking : In many of Carlini-Wagner papers, they showed that some detection methods become weak by simply increasing the confidence parameter (kappa) in the CW attack. The three attacks considered in this work, FGSM, BIM, and MIM are all L_infinity attacks, which are known to introduce unnecessary noises due to the definition of L_infinitiy norm. On the other hand, CW attack is a strong L2 attack and it also offers a way of tuning confidence of the adversarial example. In addition, a variant of CW L2 attack, called Elastic-Net attack https://arxiv.org/abs/1709.04114, is able to generate L1-norm based adversarial examples that can bypass many detection methods. Without the results of attack performance vs different confidence levels against strong L1 and L2 attacks, the detection performance is less convincing.

2. Lack of comparison to existing works - there are several detection works that already used uncertainty in detection. A representative paper is MagNet https://arxiv.org/abs/1705.09064 . MagNet paper showed that detection against FGSM/BIM is easy (even without adversarial training), and shows some level of robustness against CW L2 attack when the attacker is unaware of the detection. Later on, MagNet has been bypassed if the detection is known to the adversary https://arxiv.org/abs/1711.08478. Since MagNet and this paper have similar detection methodology using uncertainty, and the detection performance seems similar, the authors are suggested to include MagNet for comparison.

3. The objective of adaptive adversarial attack is unclear - inspecting how MagNet's detection performance is degraded when the attacker knows the detection mechanism https://arxiv.org/abs/1711.08478, the authors should do an adaptive attack that directly includes eqn (8) as one of the attack loss term, rather than using the KL term. In addition, if there is randomness in calculating the MI term for adaptive attacks, then averaged gradients over randomness should be used in adaptive attacks. Lastly, CW L2/EAD L1 attacks with an additional loss term using (8) should be compared.

4. The white-box attacks in Fig. 2 (b) to (c) seem to be quite weak - not be able to reach 100% success rate (saturates around 90%) when using BIM and MIM on the undefended model (DNN) with large attack strength. This might suggest some potential programming errors or incorrect attack implementation.

5. What black-box attack is implemented in this work? It's not clear what kind of black-box attack is implemented in this paper: is it transfer attack? score-based black-box attack? or decision-based black-box attack? Can the proposed method be robust to these three different settings?

6. This paper heavily relies on the work in  (Malinin and Gales 2018), and basically treats adversarial input detection as an out-of-distribution detection problem. Please emphasize the major differences and differentiate the contributions between these two works.

7. In Fig. 2, it seems that adversarial training with FGSM is actually the key factor that makes the detection work (by comparing PN vs PN-ADV in (b) and (c)). To justify the utility of the proposed metric in detection adversarial inputs, the authors are suggested to run MCDP on FGSM-trained model and compare the performance with PN-ADV.

---

### Official Review · AnonReviewer2 · 2018-11-03
**Interesting problem, but methodology and results are not sufficiently convincing or novel**

**Rating:** 4
**Confidence:** 4

**Review:**

Summary:
The authors propose a new method to detect adversarial attacks (examples). This approach relies on prior networks to estimate uncertainty in model predictions. Prior-networks are then trained to identify out-of-distribution inputs, and thereby used to detect adversarial examples. The authors evaluate their methodology on CIFAR-10 in different white-box and blackbox settings.

This work addresses an important question - detecting adversarial examples. Since it may not always be possible to build models that are completely robust in their predictions, detecting adversarial examples and/or identifying points where the model is uncertain is important. However, I am not convinced by the specific methodology as well as the proposed evaluation.

Detailed comments:

- This work is largely based on the recent work by Malinin and Gales, 2018, where prior networks are developed as a scheme to identify out-of-distribution inputs. As a result, the authors rely fundamentally on the assumption that adversarial examples lie off-the data manifold. There has been no convincing evidence for this hypothesis in the literature thus far. Adversarial examples are also likely to be on the data manifold, but form a small enough set that it doesn’t affect standard generalization. But because of the high-dimensional input space, a member of this small set is still close to every “natural” data point.

- I do not find the specific choice of attacks the authors consider convincing. (These being the attacks studied in Smith & Gal, 2018 does not seem to be a sufficient explanation). Specifically, the authors should evaluate on stronger Linf attacks such as PGD [Madry et al., 2017]. Further, it seems that the authors consider Linf eps around 80. These values seem extremely large given that eps=32 (possibly even > 16) causes perceptible changes in the images. Did the authors look at the adversarial examples created for these large eps values?

- The authors should include evaluation on a robust DNN (for example PGD trained VGG network) in the comparison. I believe that the joint success rate for this robust model will already be comparable to the proposed approach.

- I am not convinced by the attack that the authors provide for the setting where the detection scheme is known. This attack seems similar to the approach studied in Carlini and Wagner, 2017 (Perfect-Knowledge Attack Evaluation) which was insufficient to break the randomization defense. Why did the authors not try something along the lines of the attack in Carlini and Wagner, 2017 (Looking deeper) that actually broke the aforementioned defense? Specifically, trying to find adversarial examples that have low uncertainty as predicted by the prior networks. The uncertainty loss -- minimizing KL between p_in(\pi|x_adv) and p(\pi|x_adv, \theta) -- could be added to cross entropy loss.

- In Section 4.2, how do the authors generate black box attacks? If they are white box attacks on models trained with a different seed (as in Section 4.1) the results in 4.2 are surprising. Carlini and Wagner, 2017 found white-box attacks for randomization schemes transferrable and as per my understanding, this should be reflected in Fig 3, at least for prior work.

- I am confused by the authors comment - “Figure 3c shows that in almost 100% of cases the attack yields the target class.” The joint success rate being lower than the success rate should convey that adversarial examples couldn’t be found in many of these cases. What was the value of the epsilon that was used in these plots?

Quality, Novelty and Significance:

The paper is written well, but clarity about the evaluation procedures is lacking in the main manuscript. I am also not convinced by the rigor of the evaluation of their detection methodology. Specifically: (1) they do not consider state-of-the-art attack models such as PGD and (2) the scheme they propose for a perfect knowledge attack seems insufficient. While the paper asks an important question, I do not find the results sufficiently novel or convincing. More broadly, I find the idea of using a secondary network to detect adversarial examples somewhat tenuous as it should be fairly easy for an adversary to break this other network as well.

---

### Official Review · AnonReviewer1 · 2018-11-07
**Several inconsistencies**

**Rating:** 3
**Confidence:** 4

**Review:**

This paper proposes a new detection method for adversarial examples, based on a prior network, which gives an uncertainty estimate for the network's predictions.

The idea is interesting and the writing is clear. However, I have several major concerns. A major one of these is that the paper considers "detection of adversarial attacks" to mean detecting adaptive and non-adaptive attacks, while the latter are (a) unrealistic and (b) a heavily explored problem, with solutions ranging from clustering activations, to denoising the image via projection (in particular, once can use any of the circumvented ICLR 2018 defenses which all work in the non-adaptive sense, and check the prediction of the denoised image vs the original). Thus, the paper should focus on the regime of adaptive attacks. Within this regime:

Motivation:
- This work seems to suggest that dropout-based detection mechanisms are particularly successful. While Carlini & Wagner finds that the required distortion (using a specific attack) increases with randomization, the detection methods which used dropout were still completely circumvented in this paper.

- The claim that adversarial examples are "points off of the data manifold" is relatively unmotivated, and is not really justified. Justification for this point is needed, as it forms the entire justification for using Prior Networks.

- Detecting adversarial examples is not the same problem to detecting out-of-distribution samples, and the writing of the paper should be changed to reflect this more.

Evaluation:
- 100 iterations is not nearly enough for a randomization-based or gradient masking defense, so the attacks should be run for much longer. In particular, some of the success rate lines appear to be growing at iteration 100.

- There is no comparison to any other method (in particular, just doing robust prediction via Madry et al or something similar); this should be added to contextualize the work.

- The term "black-box" attacks can take on many meanings/threat models. The threat models in the paper need to be more well-defined, and in particular "black-box attacks" should be more accurately defined. If black-box attack refers to query-based attacks, the success rate should be equal to those of white-box attacks (or very close to it), as then the attack can just estimate the gradient through the classifier via queries.

- The fact that the attacks do not reach 100% on the unprotected classifier is concerning, and illustrates the need for stronger attacks.

Smaller comments:
Page 1: Abstract: Line 4: missing , at the end of the line
Page 1: Abstract: Line 5: “However, system can“ missing a before system
Page 1: Abstract: Line 10: “have been shown” should be “has” instead of “have”
Page 1: Abstract: Line 13: “In this work” missing a , after
Page 1: Last paragraph: Line 2: “investigate” missing an s and should be “investigates’
Page 2: Section 2: Line 5: “in other words” missing a , after
Page 2: Section 2: Line 8: “In order to capture distributional uncertainty”  missing a , after
Page 2: Last paragraph: Line 2: “Typically” missing a , after
Page 3: Paragraph 2: Line 1: “In practice, however, for deep,” no need for the last ,
Page 3: Section 2.2: paragraph 1: Line 2: “a Prior Network p(π|x∗; θˆ), “ no need for the last ,
Page 3: Section 2.2: paragraph 1: Line 3: “In this work“ missing a  , after
Page 3: second last paragraph: Line 1: refers to figure as fig and figure (not consistent)
Page 3: second last paragraph: Line 3: “uncertainty due severe class“ missing “to” before “severe”
Page 4: Paragraph 1: Line 2: “to chose” should be “to choose”
Page 4: Paragraph 2: Line 1: “Given a trained Prior Network“ missing a , after
Page 4: Paragraph 2: two extra ,
Page 6: paragraph 1: Last line: 5 needs to be written in words and same for 10 in the next paragraph
Page 7: section 4.2: paragraph 3: “For prior networks” need a , after
Page 8: Paragraph 1: Line 6: “and and”
Page 8: Conclusion: Line 4: “In section 4.2” needs a , after
Page 8: Conclusion: Line 7: “it difficult” missing “is”
Page 8: Conclusion: Line 9: “is appropriate” should be “if” instead of “is”

---

### Public Comment · (anonymous) · 2018-10-18
**Misleading maybe?**

I find this remark slightly misleading:

"(Carlini & Wagner, 2017) singles out detection of adversarial attacks using uncertainty measures derived from Monte-Carlo dropout as being successful."

The distortion increases by a bit, but it is still imperceptible (as the paper suggests) -- successful might be a strong term. The defense increases the required distortion by an imperceptible amount.

Also, why are they no tests against the CW attack itself? And, the white-box version where CW uses information about the defense (as done in the Carlini, Wagner paper cited)?

Also curious, why just one dataset and why only gradient based attacks? I would have really liked to see some experiments with gradient-free attacks like NES & optimization based attacks.

---

> ### Author Response · Authors · 2018-10-29
> **We will address your comments.**
>
> Thank you for your comments.
>
> We agree that we have overstated the results in (Carlini & Wagner, 2017) and will address this once editing the paper becomes possible.
>
> We chose to investigate detection of FGSM, BIM and MIM attacks because those were the attacks considered in the work of (Smith & Gal 2018) which defined the baseline approaches considered in our work. However, we agree that it makes sense to evaluate against the C&W attack in all threat models (White/Black box with/without knowledge of defence) for completeness. These experiments will be run once editing of the paper becomes possible.
>
> We evaluated Prior Networks on MNIST, SVHN and CIFAR-10. However, we chose to report results only on CIFAR-10 because it was the most interesting dataset and because results on MNIST and SVHN were similar to results on CIFAR-10. We can include our results on MNIST and SVHN once editing of the paper becomes possible.
>
> Best Regards,
> Authors of Paper 393

---

### Public Comment · (anonymous) · 2018-10-23
**Why cap iterations at 100?**

In Figure 3 we can see that the attack against PN-ADV is still improving at 100 iterations. What happens if this becomes 1000?

---

> ### Author Response · Authors · 2018-10-25
> **Will run attacks for 1000 iterations**
>
> Thank you for your comment.
>
> The goal of Figure 3 was to show that attacking PN-ADV is computationally expensive. While we considered 100 iterations to already be expensive, we will run the attack for 1000 iterations and report the results.
>
> Best Regards,
> Authors of Paper 393

---

### Meta-Review · Area_Chair1 · 2018-12-15
**serious issues with evaluation, some conceptual confusion**

**Confidence:** 5
**Recommendation:** Reject

**Metareview:**

This paper addresses an important topic and was generally well-written. However, reviewers pointed out serious issues with the evaluation (using weak or poorly chosen attacks), and some conceptual confusions (e.g. conflating adversarial examples with out-of-distribution examples, unsubstantiated claim that adversarial examples lie off the data manifold).